# Characteristics of Respiratory Syncytial Virus versus Influenza Infection in Hospitalized Patients of Peru: A Retrospective Observational Study

**DOI:** 10.3390/tropicalmed7100317

**Published:** 2022-10-19

**Authors:** Max Carlos Ramírez-Soto, Gutia Ortega-Cáceres, Jose Garay-Uribe

**Affiliations:** 1Centro de Investigación en Salud Pública, Facultad de Medicina Humana, Universidad San Martín de Porres, Lima 15011, Peru; 2Facultad de Ciencias de la Salud, Universidad Tecnológica del Peru, Lima 15046, Peru; 3Facultad de Ciencias de la Salud, Universidad Nacional Daniel Alcides Carrión, Pasco 19001, Peru

**Keywords:** respiratory syncytial virus, influenza, hospitalization, respiratory acute infection, Peru

## Abstract

Respiratory syncytial virus (RSV) and influenza infections are important causes of respiratory illness associated with hospitalizations in children in Peru; however, comparisons of RSV and influenza hospitalization across all age groups are not available in Peru. Therefore, we conducted an observational, retrospective study between May 2015 and October 2021 using hospitalization from RSV and influenza infection data obtained from SUSALUD (open data) in Peru to compare the baseline characteristics of sex, age, region, and infection type. For the study, 2696 RSV-infected and 1563 influenza-infected hospitalized patients from different age groups were included. Most hospitalizations from RSV infection and the influenza virus occurred in children <5 years of age (86.1% vs. 32.2%, respectively). Compared with influenza infection, RSV infection was less likely to occur in individuals ≥5 years of age (adjusted odds ratio (aOR) = 0.07; 95% confidence interval (CI), 0.06–0.08; *p* < 0.0001; compared to <5 years of age), and more likely to occur in highlands (aOR = 1.75; 95% CI, 1.46–2.07; *p* < 0.0001, compared to coast region), and jungle region (aOR = 1.75; 95% CI, 1.27–2.41; *p* = 0.001, compared to coast region). Among the respiratory complications, RSV pneumonia was less likely to occur between different age groups (aOR = 0.29; 95% CI, 0.22–0.37; *p* < 0.0001, compared to <5 years of age), compared with influenza pneumonia. These findings on the RSV-hospitalization and its complications are helpful for health services planning and may increase awareness of the Peruvian population’s RSV and influenza disease burden.

## 1. Introduction

Among respiratory virus infections, respiratory syncytial virus (RSV) and influenza infections are substantial causes of global morbidity and mortality [1,2,3]. RSV is a common cause of childhood acute lower respiratory infection (ALRI) and a major cause of hospital admissions in children aged 0–60 months. In 2015, it was estimated globally that about 3.2 million (2.7–3.8 million) hospital admissions were caused by RSV [4], while in 2019, there were about 3.6 million (2.9–4.6 million) RSV-associated ALRI hospital admissions in children aged 0–60 months, especially in low- and middle-income countries (LMICs) [5]. On the other hand, in 2018, among children under 5 years globally, there were an estimated 870,000 (543,000–1,415,000) influenza-virus-associated ALRI hospital admissions, also in LMICs. These hospitalization rates were prevalent in children <5 years of age and in adults ≥65 years [6,7].

In Peru, RSV infection is a common etiology of acute respiratory infections in children from the coast, whereas in rural communities it is associated with higher morbidity than ALRI due to other viruses [8,9,10,11,12]. Another study reveals that the incidence rate of RSV infections during the first year of life was 5.7 episodes/child-years in Lima [13]. In 2017, an influenza infection hospitalization rate of 36.8 (95% confidence interval (CI), 13.0–96.6) per 100,000 for all ages in Peru was estimated [6,7]. Hospitalizations between 2010 and 2014 were prevalent in children <5 years of age (103–259 per 100,000 people) and in adults ≥65 years (36–248 per 100,000 people), with hospitalizations for individuals aged 5–64 years old being less prevalent (21–46 per 100,000 people) [14]. High variability in the influenza burden has also been reported across ecologically distinct Peruvian regions [15]. These hospitalization rates from RSV and influenza infections result in a substantial disease burden on Peruvian health-care services. Despite this, studies on RSV- and influenza-associated hospitalization have been limited to the pediatric population, with estimates from a very limited number of participants and from several years ago, and the national burden of hospitalization across age groups, by region, and by infection type are not available. It is also important to note that several international studies show differences in clinical and epidemiological characteristics between RSV and influenza in both children and adults, but this has not been reported in Peru [16,17,18,19,20]. Understanding the burden of RSV- and influenza-associated hospitalizations would assist in the formulation of policies and would help to make informed evidence-based decisions and planning intervention strategies to limit the spread of these diseases. In addition, these estimates can provide much needed information to effectively communicate illness risk in a population and to estimate the illness costs that could be prevented.

To determine the epidemiologic pattern of RSV versus influenza and to conduct a current comparison of their characteristics, we compared the epidemiological and clinical characteristics of patients hospitalized with RSV and influenza from different age groups in Peru from May 2015 to October 2021.

## 2. Materials and Methods

### 2.1. Study Design and Population

This observational, retrospective, cross-sectional study was conducted with hospitalized patients from Peruvian regions and departments. There are three first-level administrative divisions of Peru (coast, highlands, and jungle). These regions are divided into 24 departments and a constitutional province. Hospital records for all patients admitted to an acute care hospital for a respiratory condition between May 2015, and October 2021, were extracted from the hospitalization records available on the website of the National Superintendence of Health (SUSALUD) [21]. The hospitalization records include data from the entire Peruvian health system: Ministry of Health, Regional Health Directorates, Social Security (ESSALUD), Armed Forces and Police Health services, and the private sector.

### 2.2. Source of Data

We used the International Classification of Disease, 10th Modification, (ICD-10) for coding diagnoses. Hospitalizations with a respiratory virus infection were classified into two viral groups using the ICD-10 revision code: influenza (J09.X, J10.0, J10.1, J10.8, J11.0, J11.1, and J11.8) and RSV (J12.1, J20.5, and J21.0). The diagnostic test results (polymerase chain reaction (PCR) tests) and co-infections with other ARVIs for the RSV and influenza cases were not available. All patients meeting the severe acute respiratory infections (SARI) case definition were eligible for inclusion.

We obtained the total number of patients hospitalized with RSV versus influenza infections from the hospitalization records of 315 care centers. Of each hospitalized patient admitted at the selected hospitals, age, gender, dates of admission and discharge, location of residence, and infection type were recorded. Other information on the RSV or influenza hospitalizations is not available in the data of SUSALUD.

### 2.3. Statistical Analyses

Frequency tables were generated to describe the characteristics of patients hospitalized from influenza and the RSV by sex, age (<5, 5–24, 25–44, 45–64, ≥65, and ≥5 years of age), region, department, infection type, month, and year. Chi-square tests were also used to assess for statistically significant differences in individuals aged <5 and ≥5 years during the entire study period with 95% confidence interval (CI).

We compared the demographic and clinical characteristics of RSV- versus influenza-hospitalized patients (by sex, age, region, and infection type) using bivariate and multivariate analysis with odds ratio (OR) with 95% CI. For the multivariate analysis, logistic regression was used to estimate the OR (adjusted odds ratio (aOR)) associated with the impact of all covariates (including age, sex and region) in the potential differences between the patients infected with the RSV vs. patients infected with influenza. The significant variables found during univariate analysis were taken as covariates in the multivariable regression model. For comparisons in age between the groups, covariates in the multivariable regression model included the sex and region. For comparisons in the sex between the groups, covariates included the age and region. For comparisons in hospitalizations region between the groups, covariates included the sex and age. *p*-values < 0.05 were considered significant. Analyses were performed using StataSE software version 17.0 (SPSS, Inc., Chicago, IL, USA).

### 2.4. Ethics Statements

This study did not involve human participants and hospitalizations data were obtained from SUSALUD. All data are public and available with full open access; therefore, an informed patient consent was not required. These data are anonymous and are published as part of routine surveillance. Therefore, this study was exempt from review by an Institutional Review Board.

## 3. Results

During the study period, 2696 patients hospitalized with the RSV and 1563 patients hospitalized with influenza were included. The distribution of demographic characteristics, and disease type, in patients hospitalized with the RSV or with influenza is presented in Table 1. Most hospitalizations from RSV infection and the influenza virus occurred in children <5 years of age (86.1% vs. 32.2%, respectively). There was a higher proportion of hospitalizations from the RSV in males than in females (55.2% vs. 44.8%, respectively), while the proportions of hospitalizations from the influenza virus were similar in males and females (49.0% vs. 51.0%, respectively). Most hospitalizations from RSV infection and the influenza virus occurred in 2018 (25.4% vs. 27.1%, respectively). Most hospitalizations from RSV infection and the influenza virus were registered in the coast region (62.5% vs. 70.1%, respectively), mainly in the Lima department (43.6% vs. 49.3%, respectively). Bronchiolitis was most frequent among hospitalizations from the RSV (80.2%), whereas pneumonia was more frequent in hospitalizations from influenza (52.4%) (Table 1).

### 3.1. RSV and Influenza Hospitalization Seasonal Trends

Hospitalizations from the RSV and the influenza virus were detected throughout the year with peak periods observed during May to September (Figure 1). There was a decrease in RSV and influenza hospitalizations during March 2020 through October 2021 (Figure 1). RSV-associated hospitalizations were prevalent in children under 5 years old, with a total of 307 cases in May (Figure 2A), while influenza-associated hospitalizations were prevalent in children under 5 years old and in adults between 45 and 64 years old (Figure 2B).

Compared with influenza-associated hospitalizations, the number of RSV-associated hospitalizations in children <5 years of age was significantly higher than the number of hospitalizations in individuals ≥5 years of age during the entire study period (*p* < 0.05) (Table 2).

### 3.2. Comparison of the Demographic and Clinical Characteristics of RSV- and Influenza-Hospitalized Patients

Compared with influenza-hospitalized patients, RSV-hospitalized patients were less likely to be older (5–24 years (aOR = 0.12; 95% CI, 0.10–0.16; *p* < 0.0001), 25–44 years (aOR = 0.05; 95% CI, 0.04–0.07; *p* < 0.0001), 45–64 years (aOR = 0.05; 95% CI, 0.04–0.07; *p* < 0.0001), and ≥65 years (aOR = 0.07; 95% CI, 0.05–0.09; *p* < 0.0001)), compared to <5 years of age (Table 3). The individuals ≥5 years old (aOR = 0.07; 95% CI, 0.06–0.08; *p* < 0.0001) and female sex (aOR = 0.82; 95% CI, 0.70–0.95; *p* = 0.011) were less likely to be hospitalized by RSV than for influenza, compared with children <5 years old. RSV hospitalizations were more likely to occur in patients of the highlands (aOR = 1.75; 95% CI, 1.46–2.07; *p* < 0.0001), and jungle region (aOR = 1.75; 95% CI, 1.27–2.41; *p* = 0.001), compared to coast region (Table 3).

### 3.3. Comparison of the Complications of RSV-Hospitalized Patients

Among the RSV-hospitalized patients, compared to pneumonia or bronchitis, RSV bronchiolitis was less likely to occur in individuals of 5 to ≥65 years of age, and was more likely to occur in the highlands or jungle. RSV pneumonia was more likely to occur in patients ≥65 years of age (aOR = 3.48; 95% CI, 1.31–9.23; *p* < 0.05, compared with children <5 years old), compared to bronchitis (Table 4).

### 3.4. Comparison of the Complications of Influenza-Hospitalized Patients

Among the influenza-hospitalized patients, pneumonia was less likely to occur in patients of 5–24 years (aOR = 0.40; 95% CI, 0.29–0.56; *p* < 0.0001; compared to other complications), and 25–44 years (aOR = 0.55; 95% CI, 0.40–0.76; *p* < 0.0001; compared to other complications), and was more likely to occur in ≥65 years (aOR = 2.56; 95% CI, 1.88–3.50; *p* < 0.0001; compared to other complications) compared with children <5 years old. Sex and region were not associated with pneumonia in influenza-hospitalized patients (Table 5).

### 3.5. Comparison of the Pneumonia in RSV- and Influenza-Hospitalized Patients

Compared with influenza-hospitalized patients, pneumonia in RSV-hospitalized patients had a low probability of occurring in individuals of 5–24 years (aOR = 0.32; 95% CI, 0.19–0.52; *p* < 0.0001), 25–44 years (aOR = 0.35; 95% CI, 0.22–0.55; *p* < 0.0001), 45–64 years (aOR = 0.23; 95% CI, 0.16–0.34; *p* < 0.0001), and ≥65 years (aOR = 0.30; 95% CI, 0.22–0.42; *p* < 0.0001). Sex and region were not associated with pneumonia in RSV-hospitalized patients (Table 6).

## 4. Discussion

This retrospective study demonstrates that the RSV is an important cause of serious illness among children, and less common in older age groups, comparable or more severe than that caused by influenza. Therefore, the RSV is primarily recognized as a pediatric pathogen. To our knowledge, this is the first Peruvian study to describe the largest and most recent population hospitalized with RSV or influenza of different age groups.

In our study, a greater number of hospitalizations due to RSV than influenza was evidenced in children under 5 years of age, with seasonal patterns between May and July for RSV, and June and July for influenza, with peaks in 2018 for both infections. Just as the global maps of monthly virus activity for RSV and influenza infections show the distinct seasonal latitudinal patterns for each virus in terms of both timing and duration of epidemics [22], our findings show that the hospitalization seasonal patterns of RSV and influenza infections are different in terms of timing. According to the Ministry of Health of Peru, between 2015 and 2019 there was a usual seasonal pattern in the new cases of pneumonia; these findings could help explain the high rate of hospitalizations from RSV and influenza infections and the seasonal patterns found in our study [23]. However, in 2020, there was a decrease in hospitalizations from RSV and influenza infections. This is probably due to the health establishments in 2020 in Peru being overloaded with cases of COVID-19, and the surveillance of other diseases, including SARI, being lowered, resulting in a decrease in hospitalizations from RSV and influenza infections. Another possible reason for this is that non-pharmacological interventions used to reduce the spreading of COVID-19 (as social distancing measures and continuous use of face masks) might have reduced the incidence, transmission, and hospitalizations of respiratory virus infections, as has been reported in China, Italy and Australia [24,25,26,27]. On the other hand, the highest numbers of RSV- and influenza-infection hospitalizations were concentrated on the coast, and especially in the Lima department (capital of Peru). This occurred as approximately a third of the Peruvian population lives in Lima, and this city presents a greater hospital supply, with a greater attendance capacity.

Consistent with earlier reports, we found that children <5 years of age hospitalized with RSV infection were more frequent than those hospitalized with influenza virus infection [16,17,18]. Higher rates of RSV-associated respiratory hospitalizations among children aged <5 years have been reported in Peru [13]. In addition, RSV infection has been reported with more frequency than influenza and other respiratory viruses [8,9,12]. These findings reveal that RSV infection may result in a more substantial burden on hospitalization among Peruvian children than influenza. Consistent with international reports, we also found RSV-hospitalized patients were less likely than those with influenza virus to be individuals of 5–24 years, 25–44 years, 45–64 years, and ≥65 years, compared to <5 years of age [16,17]. In contrast, an American study suggested that RSV infection may result in greater morbidity and mortality among older hospitalized adults than influenza [19]. To our knowledge, the estimated rates, and characteristics, of RSV-associated SARI hospitalization among Peruvian children aged ≥5 years, adults, and older adults have not been reported in the Peruvian literature; therefore, although the number of RSV hospitalizations in adults was lower than influenza, there has been a substantial number of RSV hospitalizations during the study period. The high probability of RSV hospitalizations in adults and older adults could be related to some chronic underlying conditions, as previously reported in the United States and France [19,20]. Therefore, more studies are needed in Peru to test this hypothesis. On the other hand, a previous report revealed a high rate of hospitalizations for influenza in all age groups in Peru and other Latin American countries [8].

In the RSV hospitalizations group, RSV bronchiolitis was less likely than pneumonia or bronchitis of occur in 5 to ≥65 years age groups, compared to <5 years age group. In contrast, pneumonia in the RSV-hospitalized patients’ group was more likely than bronchitis to occur in patients of ≥65 years of age, compared to <5 years of age. In contrast, a study in hospitalized young adults found that asthma prevalence was high after RSV bronchiolitis [28]. On the other hand, as in several studies [7,19,29], we found that pneumonia was the most common complication in patients ≥65 years with influenza infection, compared to the <5 years age group. As well as pneumonia in RSV-hospitalized patients being less common than influenza, our findings revealed that for RSV-hospitalized patients, the odds of pneumonia were lower in the older age groups, compared with influenza-hospitalized patients. A previous study reported that pneumonia, acute bronchitis, and bronchiolitis were associated with over-hospitalization in children aged 5 and younger with RSV or influenza infections [17]. However, another study in Southern California reported the highest incidence of pneumonia as a complication of RSV infection occurring in adults, especially those with higher use of systemic and inhaled antibiotics and corticosteroids [19]. In addition, in adults of Thailand, it was found that RSV is a less common cause of adult hospitalization than influenza, but the pulmonary and cardiovascular complications, and mortality, are similar [29]. Although there are no studies in Peru to discuss the association between age and bronchiolitis and pneumonia in RSV-hospitalized patients, this association could be related to the health status of patients or the presence of comorbidities, especially in older adults, as has been found in reports from other countries [19,29].

Our study has several limitations. First, the retrospective design, as the completeness of the hospital records used for the retrospective record review could not be verified. Second, cases were defined based on the ICD-10, as diagnostic test results and co-infection with other ARVIs for the RSV and influenza cases were not available (PCR tests). This could result in an overestimation in the comparisons between the odds ratio of risk in the two groups, resulting in a possible bias. Third, in this study we included data from 315 care centers, and therefore, our results are limited to these care centers and not for all of Peru. Despite these limitations, a strength of this study is the large patient sample drawn from a hospitalization population of different age groups receiving care at multiple medical centers, whereas most studies on RSV- and influenza-associated hospitalizations have been limited to the pediatric population, and as such, have a very limited number of participants, and furthermore, comparisons of RSV and influenza hospitalizations across age groups, by region and complication type, are not available.

## 5. Conclusions

Hospitalization associated with RSV infection and its complications such as bronchiolitis and pneumonia are less likely in occur in older age groups than in children aged <5 years, compared to influenza virus infection. Our findings have important implications for public health. First, this knowledge is particularly important for geographically large regions such as Lima, where great variations in the activity of both viruses might exist. Second, this information is helpful for health service planning and for the administration of health resources, especially when viruses co-circulate and impose pressure on hospital beds. Third, if an RSV vaccine should become available, this information could be used by health policymakers to consider vaccine introduction, which would prevent a substantial number of hospitalizations associated with RSV, especially in children aged <5 years and in some older age groups.

## Figures and Tables

**Figure 1 tropicalmed-07-00317-f001:**
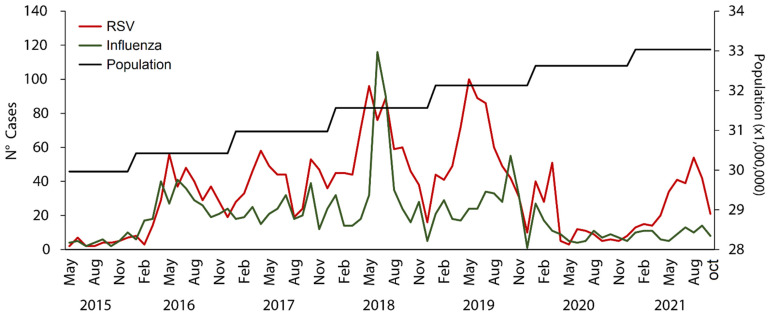
Monthly hospitalizations from the RSV and the influenza virus in Peru, seasons from 2015 to 2021.

**Figure 2 tropicalmed-07-00317-f002:**
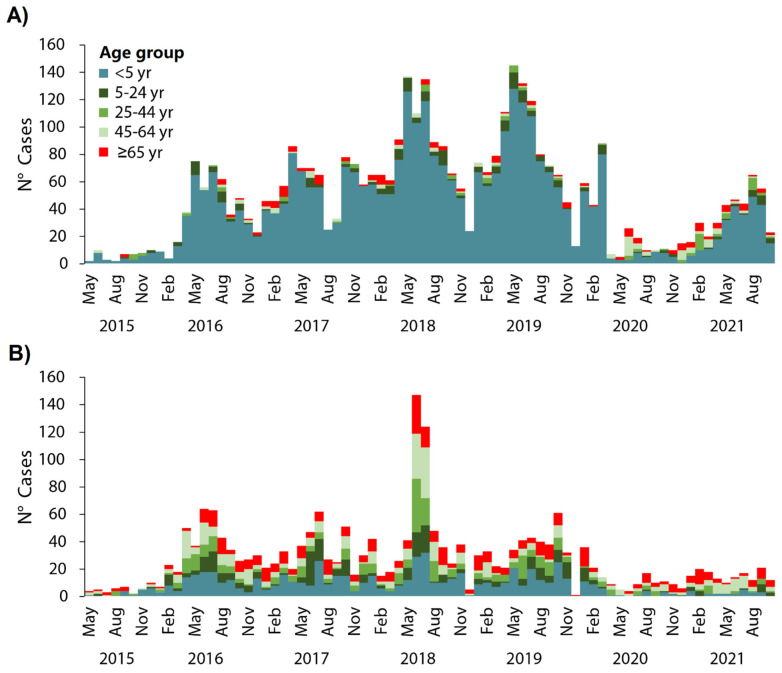
Monthly hospitalizations from the RSV (**A**) and the influenza virus (**B**) by age group in Peru, seasons from 2015 to 2021.

**Table 1 tropicalmed-07-00317-t001:** Cases of respiratory syncytial virus and influenza associated with hospitalization in Peru, 2015–2021.

	RSV-Associated Hospitalization			Influenza-Associated Hospitalization		
	Male, *n* (%)	Female, *n* (%)	Total, *n* (%)	Male, *n* (%)	Female, *n* (%)	Total, *n* (%)
**Age group (in y)**						
<5	1276 (85.6)	1046 (86.7)	2322 (86.1)	273 (35.7)	231 (28.9)	504 (32.2)
5–24	82 (5.5)	53 (4.4)	135 (5.0)	106 (13.9)	115 (14.4)	221 (14.1)
25–44	36(2.4)	26 (2.2)	62 (2.3)	105 (13.7)	129 (16.2)	234 (15.0)
45–64	44 (3.0)	29 (2.4)	73 (2.7)	147 (19.2)	150 (18.8)	297 (19.0)
≥65	52 (3.5)	52 (4.3)	104 (3.9)	134 (15.7)	173 (21.7)	307 (19.6)
All ages	1490 (55.2)	1206 (44.8)	2696 (100)	765 (49.0)	798 (51.0)	1563 (100)
**Year**						
2015	18 (1.2)	15 (1.2)	33 (1.2)	18 (2.4)	20 (2.5)	38 (2.4)
2016	186 (12.5)	162 (13.4)	348 (12.9)	136 (17.8)	168 (21.1)	304 (19.4)
2017	254 (17.0)	227 (18.8)	481 (17.8)	122 (15.9)	145 (18.2)	267 (17.1)
2018	387 (26.0)	298 (24.7)	685 (25.4)	218 (28.5)	206 (25.8)	424 (27.1)
2019	374 (25.1)	299 (24.8)	673 (25.0)	157 (20.5)	159 (19.9)	316 (20.2)
2020	104 (7.0)	79 (6.6)	183 (6.8)	71 (9.3)	46 (5.8)	117 (7.5)
2021	167 (11.2)	126 (10.4)	293 (10.9)	43 (5.6)	54 (6.8)	97 (6.2)
**Region**						
Coast	941 (63.2)	744 (61.7)	1685 (62.5)	549 (71.8)	547 (68.5)	1096 (70.1)
Highlands	450 (30.2)	378 (31.3)	828 (30.7)	176 (23.0)	200 (25.1)	376 (24.1)
Jungle	99 (6.6)	84 (7.0)	183 (6.8)	40 (5.2)	51 (6.4)	91 (5.8)
**Department**						
Amazonas	2 (0.1)	3 (0.2)	5 (0.2)	5 (0.7)	11 (1.4)	16 (1.0)
Ancash	53 (3.6)	47 (3.9)	100 (3.7)	13 (1.7)	5 (0.6)	18 (1.2)
Apurimac	14 (0.9)	14 (1.2)	28 (1.0)	6 (0.8)	17 (2.1)	23 (1.5)
Arequipa	31 (2.1)	26 (2.2)	57 (2.1)	71 (9.3)	81 (10.2)	152 (9.7)
Ayacucho	70 (4.7)	68 (5.6)	138 (5.1)	9 (1.2)	8 (1.0)	17 (1.1)
Cajamarca	23 (1.5)	17 (1.4)	40 (1.5)	12 (1.6)	14 (1.8)	26 (1.7)
Callao	67 (4.5)	49 (4.1)	116 (4.3)	55 (7.2)	53 (6.6)	108 (6.9)
Cusco	76 (5.1)	80 (6.6)	156 (5.8)	19 (2.5)	21 (2.6)	40 (2.6)
Huancavelica	32 (2.1)	19 (1.6)	51 (1.9)	9 (1.2)	16 (2.0)	25 (1.6)
Huanuco	14 (0.9)	16 (1.3)	30 (1.1)	6 (0.8)	4 (0.5)	10 (0.6)
Ica	29 (1.9)	23 (1.9)	52 (1.9)	10 (1.3)	12 (1.5)	22 (1.4)
Junin	59 (4.0)	39 (3.2)	98 (3.6)	14 (1.8)	14 (1.8)	28 (1.8)
La Libertad	17 (1.1)	15 (1.2)	32 (1.2)	25 (3.3)	46 (5.8)	71 (4.5)
Lambayeque	100 (6.7)	82 (6.8)	182 (6.8)	33 (4.3)	32 (4.0)	65 (4.2)
Lima	653 (43.8)	522 (43.3)	1175 (43.6)	387 (50.6)	384 (48.1)	771 (49.3)
Loreto	42 (2.8)	40 (3.3)	82 (3.0)	15 (2.0)	13 (1.6)	28 (1.8)
Madre de Dios	12 (0.8)	5 (0.4)	17 (0.6)	2 (0.3)	2 (0.3)	4 (0.3)
Moquegua	4 (0.3)	2 (0.2)	6 (0.2)	2 (0.3)	2 (0.3)	4 (0.3)
Pasco	3 (0.2)	3 (0.2)	6 (0.2)	3 (0.4)	1 (0.1)	4 (0.3)
Piura	51 (3.4)	45 (3.7)	96 (3.6)	31 (4.1)	13 (1.6)	44 (2.8)
Puno	75 (5.0)	49 (4.1)	124 (4.6)	14 (1.8)	19 (2.4)	33 (2.1)
San Martin	13 (0.9)	8 (0.7)	21 (0.8)	16 (2.1)	22 (2.8)	38 (2.4)
Tacna	11 (0.7)	2 (0.2)	13 (0.5)	4 (0.5)	3 (0.4)	7 (0.4)
Tumbes	9 (0.6)	4 (0.3)	13 (0.5)	2 (0.3)	2 (0.3)	4 (0.3)
Ucayali	30 (2.0)	28 (2.3)	58 (2.2)	2 (0.3)	3 (0.4)	5 (0.3)
**Disease Type**						
Bronchiolitis	1188 (79.7)	974 (80.8)	2162 (80.2)	0 (0.0)	0 (0.0)	0 (0.0)
Pneumonia	244 (16.4)	184 (15.3)	428 (15.9)	416 (54.4)	403 (50.5)	819 (52.4)
Bronchitis	58 (3.9)	48 (4.0)	106 (3.9)	0 (0.0)	0 (0.0)	0 (0.0)
Other complications by influenza	–	–	–	349 (45.6)	395 (49.5)	744 (47.6)

RSV, respiratory syncytial virus; y, years.

**Table 2 tropicalmed-07-00317-t002:** Annual cases of respiratory syncytial virus and influenza infections in hospitalized patients in Peru, 2015–2021.

Age Group (in y)	RSV-Associated Hospitalization		Influenza-Associated Hospitalization		*p*-Value *
	N = 2696	(%)	N = 1563	(%)	
2015					
<5	25	75.8	16	42.1	0.004
≥5	8	24.2	22	57.9	
2016					
<5	309	88.8	92	31.3	<0.0001
≥5	39	11.2	212	68.7	
2017					
<5	428	89	104	39	<0.0001
≥5	53	11	163	61	
2018					
<5	608	88.8	123	29.0	<0.0001
≥5	77	11.2	301	71.0	
2019					
<5	601	89.3	118	37.3	<0.0001
≥5	72	10.7	198	62.7	
2020					
<5	138	75.4	33	28.2	<0.0001
≥5	45	24.6	84	71.8	
2021					
<5	213	72.3	18	18.6	<0.0001
≥5	80	27.3	79	81.4	

RSV, respiratory syncytial virus; y, years. * Chi-square; *p* < 0.05 is considered significant.

**Table 3 tropicalmed-07-00317-t003:** Factors associated with the respiratory syncytial virus compared to the influenza virus among patients hospitalized in Peru, 2015–2021.

Covariate	RSV-Associated Hospitalization	Influenza-Associated Hospitalization	Univariate Analysis		Multivariate Analysis **	
	N = 2696	N = 1563	Crude OR (95% CI)	*p*-Value *	Adjusted OR (95% CI)	*p*-Value *
**Age (in y)**						
<5	2322	504	1 (ref)		1 (ref)	
5–24	135	221	0.13 (0.10–0.16)	<0.0001	0.12 (0.10–0.16)	<0.0001
25–44	62	234	0.05 (0.04–0.07)	<0.0001	0.05 (0.04–0.07)	<0.0001
45–64	73	297	0.05 (0.04–0.07)	<0.0001	0.05 (0.04–0.07)	<0.0001
≥65	104	307	0.07 (0.05–0.09)	<0.0001	0.07 (0.05–0.09)	<0.0001
**Age group (in y)**						
<5	2322	504	1 (ref)		1 (ref)	
≥5	374	1059	0.08 (0.07–0.09)	<0.0001	0.07 (0.06–0.08)	<0.0001
**Sex**						
Male	1490	765	1 (ref)		1 (ref)	
Female	1206	798	0.77 (0.68–0.87)	<0.0001	0.82 (0.70–0.95)	0.011
**Region**						
Coast	1685	1096	1 (ref)		1 (ref)	
Highlands	828	376	1.43 (1.24–1.65)	<0.0001	1.75 (1.46–2.07)	<0.0001
Jungle	183	91	1.30 (1.00–1.70)	0.045	1.75 (1.27–2.41)	0.001

RSV, respiratory syncytial virus; OR, odds ratio; CI, confidence interval; y, years. * *p* < 0.05 is considered significant. ** Adjusted for every other covariate through a multivariate logistic regression model.

**Table 4 tropicalmed-07-00317-t004:** RSV respiratory tract infections by age, sex, and region in hospitalized patients in Peru, 2015–2021.

Covariate	Bronchiolitis (n = 2162)	Pneumonia (n = 428)	Bronchitis (n = 106)	Adjusted OR (95% CI) *		
				Bronchiolitis vs. Pneumonia	Bronchiolitis vs. Bronchitis	Pneumonia vs. Bronchitis
**Age (in y)**						
<5	1993	268	61	1 (ref)	1 (ref)	1 (ref)
5–24	93	22	20	0.53 (0.32–0.86) **	0.13 (0.08–0.24) **	0.24 (0.12–0.47) **
25–44	24	30	8	0.08 (0.04–0.14) **	0.09 (0.04–0.22) **	0.97 (0.41–2.31)
45–64	20	41	12	0.04 (0.03–0.08) **	0.05 (0.02–0.10) **	0.83 (0.40–1.71)
≥65	32	67	5	0.04 (0.03–0.8) **	0.19 (0.07–0.52) **	3.48 (1.31–9.23) **
**Age group (in y)**						
<5	1993	268	61	1 (ref)	1 (ref)	1 (ref)
≥5	169	160	45	0.11 (0.09–0.15) **	0.11 (0.07–0.17) **	0.81 (0.52–1.27)
**Sex**						
Male	1188	244	58	1 (ref)	1 (ref)	1 (ref)
Female	974	184	48	1.08 (0.86–1.37)	0.94 (0.62–1.42)	0.84 (0.54–1.31)
**Region**						
Coast	1274	331	80	1 (ref)	1 (ref)	1 (ref)
Highlands	732	76	20	3.67 (2.69–5.01) **	2.32 (1.38–3.90) **	0.69 (0.38–1.27)
Jungle	156	21	6	2.53 (1.49–4.31) **	2.09 (0.86–5.06)	0.70 (0.26–1.86)

RSV, respiratory syncytial virus; OR, odds ratio; CI, confidence interval; y, years. * Adjusted for every other covariate through a multivariate logistic regression model. ** *p* < 0.05 is considered significant.

**Table 5 tropicalmed-07-00317-t005:** Influenza respiratory tract infections by age, sex, and region in hospitalized patients in Peru, 2015–2021.

Covariate	Influenza-Associated Hospitalization		Univariate Analysis		Multivariate Analysis **	
	Pneumonia (n = 819)	Other Complications (n = 744)	Crude OR (95% CI)	*p*-Value *	Adjusted OR (95% CI)	*p*-Value *
**Age (in y)**						
<5	264	240	1 (ref)		1 (ref)	
5–24	68	153	0.40 (0.28–0.56)	<0.0001	0.40 (0.29–0.56)	<0.0001
25–44	88	146	0.54 (0.39–0.75)	<0.0001	0.55 (0.40–0.76)	<0.0001
45–64	173	124	1.26 (0.94–1.69)	0.107	1.27 (0.95–1.70)	0.105
≥65	226	81	2.53 (1.86–3.45)	<0.0001	2.56 (1.88–3.50)	<0.0001
**Age group (in y)**						
<5	264	240	1 (ref)		1 (ref)	
≥5	555	504	1.00 (0.80–1.23)	0.992	1.00 (0.81–1.24)	0.963
**Sex**						
Male	416	349	1 (ref)		1 (ref)	
Female	403	395	0.85 (0.70–1.04)	0.125	0.83 (0.67–1.02)	0.079
**Region**						
Coast	570	526	1 (ref)		1 (ref)	
Highlands	207	169	1.13 (0.89–1.43)	0.307	1.08 (0.84–1.38)	0.535
Jungle	42	49	0.79 (0.51–1.21)	0.284	0.87 (0.56–1.37)	0.571

OR, odds ratio; CI, confidence interval; y, years. * *p* < 0.05 is considered significant. ** Adjusted for every other covariate through a multivariate logistic regression model.

**Table 6 tropicalmed-07-00317-t006:** Pneumonia associated with the RSV compared to the influenza virus among patients hospitalized in Peru, 2015–2021.

Covariate	RSV-Associated Hospitalization	Influenza-Associated Hospitalization	Univariate Analysis		Multivariate Analysis **	
	N = 428	N = 819	Crude OR (95% CI)	*p*-Value *	Adjusted OR (95% CI)	*p*-Value *
**Age (in y)**						
<5	268	264	1 (ref)		1 (ref)	
5–24	22	68	0.31 (0.19–0.53)	<0.0001	0.32 (0.19–0.52)	<0.0001
25–44	30	88	0.33 (0.21–0.52)	<0.0001	0.35 (0.22–0.55)	<0.0001
45–64	41	173	0.23 (0.15–0.34)	<0.0001	0.23 (0.16–0.34)	<0.0001
≥65	67	226	0.29 (021–0.40)	<0.0001	0.30 (0.22–0.42)	<0.0001
**Age group (in y)**						
<5	268	264	1 (ref)		1 (ref)	
≥5	160	555	0.28 (0.22–0.36)	<0.0001	0.29 (0.22–0.37)	<0.0001
**Sex**						
Male	244	416	1 (ref)		1 (ref)	
Female	184	403	0.77 (0.61–0.98)	0.037	0.80 (0.63–1.03)	0.090
**Region**						
Coast	331	570	1 (ref)		1 (ref)	
Highlands	76	207	0.63 (0.47–0.84)	0.002	0.85 (0.62–1.17)	0.345
Jungle	21	42	0.58 (0.50–1.47)	0.588	0.97 (0.55–1.72)	0.937

RSV, respiratory syncytial virus; OR, odds ratio; CI, confidence interval; y, years. * *p* < 0.05 is considered significant. ** Adjusted for every other covariate through a multivariate logistic regression model.

## Data Availability

The data presented in this study are publicly available at: Superintendencia Nacional de Salud (SUSALUD). Consolidated Morbidity in Hospitalization. Available online: http://datos.susalud.gob.pe/dataset/consulta-d2-consolidado-de-morbilidad-en-hospitalizacion (accessed on 30 November 2021); INEI, Peruvian population: https://www.inei.gob.pe/estadisticas/indice-tematico/population-estimates-and-projections/ (accessed on 30 November 2021).

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
