# Peer review of "Characteristics of Respiratory Syncytial Virus versus Influenza Infection in Hospitalized Patients of Peru: A Retrospective Observational Study"

_tropicalmed, 2022, doi:10.3390/tropicalmed7100317_

Round 1

Reviewer 1 Report (Previous Reviewer 2)

This is an interesting study of the morbidity from RSV and influenza in Peru among different age groups in Peru. The dataset included a representative cohort of patients that allowed comparison between the impact of these two respiratory pathogens. The study is well designed and the paper is well-written; however, there are some flaws that need to be addressed prior to publication.

Major comments:

1.       The authors included into analyzes patients with influenza which had disease codes J11, J11.1 and J11.8 (ICD-10 classifications). However, these codes are assigned to cases where the influenza virus was not identified, meaning that other respiratory pathogens could be associated with the disease symptoms similar to influenza.

2.       Figure 2 shows the sum of the cases for individual months in different years, but it would be more informative to show each year separately, since different seasons may show different trends, and the overall conclusion from this figure may be misleading. If all years have the same trends, the conclusions will be sound.

Minor comments

1.       The authors give P values equal 0.0001 in many different comparisons, however it is obvious that such precise value could not be calculated. The authors should write “p<0.0001” when the difference is highly significant instead of “p=0.0001”.

2.       Lane 19. “RSV infections were” or “RSV infection was”, please correct

3.       Lane 20. “in individuals”?

4.       Lane 21: “in highlands”?

5.       Lanes 23-25. The main message of this sentence is not clear. Does it refer to the difference in RSV pneumonia between different age groups or to the difference between RSV and influenza complications. Please revise the sentence.

6.       Lanes 41, 48, 166, 178, 187, 197, 238, 251, 255: “≥65+” – the plus sign is redundant, please replace with “≥65”.

7.       Lanes 43-44. The second part of the sentence is confusing, please rephrase.

8.       Lane 70. Better to write “Study design and population”

9.       Lane 88. “SARI” – decipher the abbreviation when first used.

10.   Lane 135: “was more frequent in hospitalizations from influenza”?

11.   Table 2 footnote. It is not clear what the CI refers to, as no confidence intervals are shown.

Author Response

Response-to-reviewers: Manuscript ID tropicalmed-1983480

We thank the Reviewer for your comments and constructive criticism, we believe that the quality of our manuscript has been significantly improved. We have revised our paper in a point-by-point manner.

Major comments:

Comment 1: The authors included into analyzes patients with influenza which had disease codes J11, J11.1 and J11.8 (ICD-10 classifications). However, these codes are assigned to cases where the influenza virus was not identified, meaning that other respiratory pathogens could be associated with the disease symptoms similar to influenza. 

Response 1: Thank you for your comments. In our study, for influenza hospitalizations, we using ICD-10 code J09.X, J10.0, J10.1, J10.8, J11.0, J11.1, and J11.8, according to the studies by Učakar et al. (ICD-10 code J12–J18), and Auvinen et al. (ICD-10 code J09-J11). We acknowledge this as a limitation. Therefore, we explain in the Discussion section (lines 272-276). 

Učakar et al. The impact of influenza and respiratory syncytial virus on hospitalizations for lower respiratory tract infections in young children: Slovenia, 2006-2011. Influenza Other Respir Viruses. 2013 Nov;7(6):1093-102. doi: 10.1111/irv.12134.

Auvinen et al. Clinical characteristics and population-based attack rates of respiratory syncytial virus versus influenza hospitalizations among adults-An observational study. Influenza Other Respir Viruses. 2022 Mar;16(2):276-288. doi: 10.1111/irv.12914.

Comment 2: Figure 2 shows the sum of the cases for individual months in different years, but it would be more informative to show each year separately, since different seasons may show different trends, and the overall conclusion from this figure may be misleading. If all years have the same trends, the conclusions will be sound.

Response 2: Thank you for your comments. We have prepared Figure 2 according to your recommendations.

Minor comments

Comment 1: The authors give P values equal 0.0001 in many different comparisons, however it is obvious that such precise value could not be calculated. The authors should write “p<0.0001” when the difference is highly significant instead of “p=0.0001”.

Response 1: Thank you for your comments. We have made the correction.

Comment 2: Lane 19. “RSV infections were” or “RSV infection was”, please correct

Response 2: Thank you for your comments. We have made the correction.

Comment 3: Lane 20. “in individuals”?

Response 3: Thank you for your comments. We have made the correction.

Comment 4: Lane 21: “in highlands”?

Response 4: Thank you for your comments. We have made the correction.

Comment 5: Lanes 23-25. The main message of this sentence is not clear. Does it refer to the difference in RSV pneumonia between different age groups or to the difference between RSV and influenza complications. Please revise the sentence.

Response 5: Thank you for your comments. We have made the correction.

Comment 6: Lanes 41, 48, 166, 178, 187, 197, 238, 251, 255: “≥65+” – the plus sign is redundant, please replace with “≥65”.

Response 6: Thank you for your comments. We have made the correction.

Comment 7: Lanes 43-44. The second part of the sentence is confusing, please rephrase.

Response 7: Thank you for your comments. We have made the correction.

Comment 8: Lane 70. Better to write “Study design and population”.

Response 8: Thank you for your comments. We have made the correction.

Comment 9: Lane 88. “SARI” – decipher the abbreviation when first used.

Response 9: Thank you for your comments. We have made the correction.

Comment 10: Lane 135: “was more frequent in hospitalizations from influenza”?

Response 10: Thank you for your comments. We have made the correction.

Comment 11: Table 2 footnote. It is not clear what the CI refers to, as no confidence intervals are shown.

Response 11: Thank you for your comments. We have made the correction of the error.

Reviewer 2 Report (Previous Reviewer 1)

The authors corrected the mistake. Odds ratio is a value from comparison which add a limitation for the conclusion. In this paper, the authors mostly compared RSV to influenza virus, so in the discussion and conclusion, the authors should clearly indicate this. such as line 236-237: it should be "RSV-hospitalized patients were less likely than influenza virus to be individuals ..."

line 184-186: the authors should add "compared to other complications".

line 250: "In the RSV hospitalizations group, RSV bronchiolitis was less likely of occur in 5 to ≥65+ years age groups. "the authors should add two limitations: 1. compared to <5 years age group; 2. compared to Pneumonia.

There have more similar conclusions need to be corrected. Please revise them.

Author Response

Response-to-reviewers: Manuscript ID tropicalmed-1983480

We thank the Reviewer for your comments and constructive criticism, we believe that the quality of our manuscript has been significantly improved. We have revised our paper in a point-by-point manner.

Comment 1: Such as line 236-237: it should be "RSV-hospitalized patients were less likely than influenza virus to be individuals ..."

Response 1: Thank you for your comments. We have made the correction.

Comment 2: line 184-186: the authors should add "compared to other complications".

Response 2: Thank you for your comments. We have made the correction.

Comment 3: line 250: "In the RSV hospitalizations group, RSV bronchiolitis was less likely of occur in 5 to ≥65+ years age groups. "the authors should add two limitations: 1. compared to <5 years age group; 2. compared to Pneumonia.

Response 3: Thank you for your comments. We have made the correction.

Comment 4: There have more similar conclusions need to be corrected.

Response 4: Thank you for your comments. We have made the corrections (see Discussion section).

Round 2

Reviewer 1 Report (Previous Reviewer 2)

The authors have adequately addressed the issues raised during the original peer review. The only comment refers to the P values: the sign "<" is given only for the P values less than 0.0001; in other cases the exact values should be presented (e.g. lane 23, Table 2 upper row and Table 3 lower rows).

Author Response

We thank the Reviewer for your comments and constructive criticism, we believe that the quality of our manuscript has been significantly improved. We have revised our paper in a point-by-point manner.

Comment: The authors have adequately addressed the issues raised during the original peer review. The only comment refers to the P values: the sign "<" is given only for the P values less than 0.0001; in other cases the exact values should be presented (e.g. lane 23, Table 2 upper row and Table 3 lower rows).

Response: Thank you for your comments. We have made the correction.

Reviewer 2 Report (Previous Reviewer 1)

I think the authors addressed my questions well and the paper is suitable for publication.

Author Response

We thank the Reviewer for your comments and constructive criticism, we believe that the quality of our manuscript has been significantly improved.

This manuscript is a resubmission of an earlier submission. The following is a list of the peer review reports and author responses from that submission.

Round 1

Reviewer 1 Report

I would like to thank for the opportunity to review this paper.

Respiratory Syncytial Virus and Influenza virus are two leading causes of respiratory diseases. This paper reported a retrospective observational study reviews the characteristics of Respiratory Syncytial Virus versus Influenza Infection in Hospitalized Patients of Peru regarding sex, age, region, and disease type based on the data from SUSALUD which covers from January 2015 to November 2021. The data of 2696 RSV-infected and 1563 influenza-infected hospitalized patients was analyzed in this study. Based on the data analysis, the authors found that most hospitalizations from RSV infection and the influenza virus occurred in children <5 years of age and claimed that RSV-hospitalized patients had a high probability of being older compared to <5 years of age and in RSV-hospitalized patients the probability of pneumonia increased with age (p<0.05) compared to the influenza-hospitalized patients. The authors also concluded that bronchiolitis caused by RSV was associated with all age groups, while pneumonia influenza was associated with the age groups of 5-24 years and 25-44 years. However, those conclusions are not well illustrated in the paper from the data reported.

Here are some questions that need to be addressed:

1.      Line 101 to 108. Methods. Please describe clearly how to do bivariate and multivariate analysis with odds ratio (OR). How is the OR calculated and what does this value mean?

2.      Figure 1: the data should cover from January 2015 to November 2021, while the Figure 1 only shows data from May 2015 to maybe Oct 2021. And the hospitalizations from the RSV and the influenza virus in 2015 and 2020 are very low. The year 2020 may be due to COVID19 (line210), however, why 2015 is also low? And whether the influence of COVID19 will affect the results in this study? And should the authors exclude the data of 2020?

3.      As the authors mentioned in Line 214 to 216, the population and hospitals distribution greatly influence the hospitalizations data. Thus, without the consideration of population, the results of regions are invalid.

4.      From table 3 to table 6, the data of group <5 years old was used as Ref. However, in the discussion, the authors sometimes did not include this group for discussion. Such as line 223-225 “we also found RSV-hospitalized patients were more likely to be individuals of 5-24 years, 25-44 years, 45-64 years, and 65+ years, compared to <5 years of age [16,17].” Also here included all groups > 5 years old. Line 237-238: “In contrast, pneumonia in RSV-hospitalized patients was more likely to occur in patients of 5-24 years of age.” Should not it be < 5 years of age?

5.      As I do not know how the odd ratio calculates, some conclusions are hard to understand for me, such as “RSV-hospitalized patients had a high probability of being older (5-24 years (adjusted odds ratio (aOR)=3.07; 95% confidence interval (CI), 1.84-5.12; p=0.000), 25-44 years (aOR=2.83; 95% CI, 1.79-4.47; p=0.000), 45-64 years (aOR=4.24; 95% CI, 2.89-6.22; p=0.000), and ≥65+ years (aOR=3.25; 95% CI, 2.34-4.52; p=0.000)) compared to <5 years of age” and “Bronchiolitis caused by RSV was associated with all age groups, while pneumonia influenza was associated with the age groups of 5-24 years and 25-44 years”. Should not pneumonia influenza be associated with the age groups of <5 and >65 as they have high hospitalizations.

6.      Line 200-201: “In our study, a greater number of hospitalizations due to RSV than influenza was evidenced in children under 5 years of age, with seasonal patterns between January and July for RSV, and May and September for influenza, with peaks in 2018 for both infections.” However, from Figure 2, it shows that May to July for RSV and June to July for influenza are the highest peaks in the year.

Author Response

Response-to-reviewers: Manuscript ID tropicalmed-1924764

We thank the Reviewer for your comments and constructive criticism, we believe that the quality of our manuscript has been significantly improved. We have revised our paper in a point-by-point manner.

Comment 1. Line 101 to 108. Methods. Please describe clearly how to do bivariate and multivariate analysis with odds ratio (OR). How is the OR calculated and what does this value mean?

Response 1. Thank you for your comments. We describe the bivariate and multivariate analysis with odds ratio (OR) on lines 105-112.

Comment 2. Figure 1: the data should cover from January 2015 to November 2021, while the Figure 1 only shows data from May 2015 to maybe Oct 2021. And the hospitalizations from the RSV and the influenza virus in 2015 and 2020 are very low. The year 2020 may be due to COVID19 (line210), however, why 2015 is also low? And whether the influence of COVID19 will affect the results in this study? And should the authors exclude the data of 2020?

Response 2: Thank you for your comments. The sentence was corrected. We included the hospital records for all patients admitted to an acute care hospital for a respiratory condition between May 2015, and October 2021. We do not consider excluding data from 2015 and 2020, because we compared the epidemiological and clinical characteristics of patients hospitalized with RSV and influenza from different age groups in Peru. We do not study the prevalence and incidence, indicators that could be affected by the base population. In addition, we include the following limitation "Third, in this study we included data from 315 care centers, therefore, our results are limited to these care centers and not for all of Peru".

Comment 3. As the authors mentioned in Line 214 to 216, the population and hospitals distribution greatly influence the hospitalizations data. Thus, without the consideration of population, the results of regions are invalid.

Response 3: Thank you for your comments. In this study we included data from 315 care centers, therefore, our results are limited to these care centers and not for all of Peru. Table 1, only summarizes the basic characteristics of the study population, including the regions of care centers that diagnosed the cases. Because of, we cannot include the region population. This has been included as a limitation of the study.

Comment 4. From table 3 to table 6, the data of group <5 years old was used as Ref. However, in the discussion, the authors sometimes did not include this group for discussion. Such as line 223-225 “we also found RSV-hospitalized patients were more likely to be individuals of 5-24 years, 25-44 years, 45-64 years, and ≥65+ years, compared to <5 years of age [16,17].” Also here included all groups > 5 years old. Line 237-238: “In contrast, pneumonia in RSV-hospitalized patients was more likely to occur in patients of 5-24 years of age.” Should not it be < 5 years of age?

Response 4: Response 6: Thank you for your comments. The sentences were corrected.

Comment 5. As I do not know how the odd ratio calculates, some conclusions are hard to understand for me, such as “RSV-hospitalized patients had a high probability of being older (5-24 years (adjusted odds ratio (aOR)=3.07; 95% confidence interval (CI), 1.84-5.12; p=0.000), 25-44 years (aOR=2.83; 95% CI, 1.79-4.47; p=0.000), 45-64 years (aOR=4.24; 95% CI, 2.89-6.22; p=0.000), and ≥65+ years (aOR=3.25; 95% CI, 2.34-4.52; p=0.000)) compared to <5 years of age” and “Bronchiolitis caused by RSV was associated with all age groups, while pneumonia influenza was associated with the age groups of 5-24 years and 25-44 years”. Should not pneumonia influenza be associated with the age groups of <5 and >65 as they have high hospitalizations.

Response 5: The bivariate (ORc; OR is the cross-product ratio) or multivariate (ORa) analyzes are performed on a reference value (Ref). This reference value (Ref.) must be a number greater than the other comparison groups.

Comment 6. Line 200-201: “In our study, a greater number of hospitalizations due to RSV than influenza was evidenced in children under 5 years of age, with seasonal patterns between January and July for RSV, and May and September for influenza, with peaks in 2018 for both infections.” However, from Figure 2, it shows that May to July for RSV and June to July for influenza are the highest peaks in the year.

Response 6: Thank you for your comments. The sentence was corrected.

Reviewer 2 Report

This study presents the results of an observational, retrospective study of RSV and influenza hospitalized cases in Peru in seasons 2015-2020. The authors compared the attack rates of these infections among different age groups, as well as other variables of this dataset. The data were obtained from the National Superintendence of Health (SUSALUD) database. The paper is well-written and presents some interesting findings. The study is scientifically sound and has some added value, however, some clarification is needed prior to publication:

1.       The authors need to clarify how the diagnoses were done (PCR tests?), and whether the patients could be co-infected with other ARVIs. Or with SARS-CoV-2 in 2020 and 2021 seasons.

2.       Lanes 24-24 please check if the phrase “pneumonia influenza” is correct.

3.       P values 0.000 look very odd, please check it carefully

4.       Lanes 127-129: The sentence states “Pneumonia was most frequent among hospitalizations from the RSV (80.2%), whereas bronchitis was in hospitalizations from the influenza virus (52.4%) (Table 1)”, whereas Table 1 shows 80.2% bronchiolitis in RSV-infected patients and 52.4% pneumonia in influenza-infected patients. Please make the text and the tables consistent

5.       Table 1. It is not clear what the term “Others influenza” means

6.       Lanes 209-211. The authors suggest that the decrease in RSV hospitalization in 2020-2021 was due to the problems in ARVI surveillance in the COVID-19 era, but there could be the real decrease in virus circulation (as it was for influenza viruses). The authors need to discuss this probability as well and find relevant references on the RSV circulation in the World in those seasons.

Author Response

Response-to-reviewers: Manuscript ID tropicalmed-1924764

We thank the Reviewer for your comments and constructive criticism, we believe that the quality of our manuscript has been significantly improved. We have revised our paper in a point-by-point manner.

Reviewer 2

Comment 1. The authors need to clarify how the diagnoses were done (PCR tests?), and whether the patients could be co-infected with other ARVIs. Or with SARS-CoV-2 in 2020 and 2021 seasons.

Response 1. Thank you for your comments. The diagnostic test results (polymerase chain reaction tests) and co-infection with other ARVIs for the RSV and influenza cases were not available (lines 88-90). We extracted the hospitalization records available on the website of the National Superintendence of Health (SUSALUD), Peru. Hospitalizations with a respiratory virus infection were classified into 2 viral groups using the ICD-10 revision code: influenza (J09.X, J10.0, J10.1, J10.8, J11.0, J11.1, and J11.8) and RSV (J12.1, J20.5, and J21.0). All patients meeting the SARI case definition were eligible for inclusion. This is a limitation of the study (Discussion section; lines 259-261). 

Comment 2. Lanes 24-24 please check if the phrase “pneumonia influenza” is correct.

Response 2: Thank you for your comments. The sentence was corrected.

Comment 3. P values 0.000 look very odd, please check it carefully

Response 3. Thank you for your comments. The P values were corrected "0.0001".

Comment 4. Lanes 127-129: The sentence states “Pneumonia was most frequent among hospitalizations from the RSV (80.2%), whereas bronchitis was in hospitalizations from the influenza virus (52.4%) (Table 1)”, whereas Table 1 shows 80.2% bronchiolitis in RSV-infected patients and 52.4% pneumonia in influenza-infected patients. Please make the text and the tables consistent.

Response 4. Thank you for your comments. The sentence was corrected.

Comment 5. Table 1. It is not clear what the term “Others influenza” means.

Response 5. Thank you for your comments. The sentence was corrected “Other complications by influenza”.

Comment 6. Lanes 209-211. The authors suggest that the decrease in RSV hospitalization in 2020-2021 was due to the problems in ARVI surveillance in the COVID-19 era, but there could be the real decrease in virus circulation (as it was for influenza viruses). The authors need to discuss this probability as well and find relevant references on the RSV circulation in the World in those seasons.

Response 6. Thank you for your comments. We discuss this probability with relevant references on the RSV circulation in the World during COVID-19 pandemic.

Round 2

Reviewer 1 Report

The authors addressed some of the questions, however, some serious issues need to be further addressed.

1.     Could you please provide the details for the calculation of odds ratio in table 2-table 6? OR = (a/c)/(b/d). The interpretation of the data by the authors should be wrong. For example, in table 3, age<5 with RSV 2322, and Flu 504; age 5-24 with RSV 135 and Flu 221. The authors calculated OR as (2322/504)/(135/221)=7.54, which is the ratio of RSV/Flu in age <5 is 7.54 fold higher to that of the age 5-24, indicating that age <5 is more vulnerable by RSV than Flu compare to age 5-24. Thus, many conclusions in this paper is not correct. Such as “RSV-hospitalized patients had a high probability of being older (5-24 years (adjusted odds ratio (aOR)=3.07; 95% confidence interval (CI), 1.84-5.12; p=0.000), 25-44 years (aOR=2.83; 95% CI, 1.79-4.47; p=0.000), 45-64 years (aOR=4.24; 95% CI, 2.89-6.22; p=0.000), and ≥65+ years (aOR=3.25; 95% CI, 2.34-4.52; p=0.000)) compared to <5 years of age” and “Bronchiolitis caused by RSV was associated with all age groups, while pneumonia influenza was associated with the age groups of 5-24 years and 25-44 years”. Some conclusions should be reversed.

2.     The calculation in table 3 may be not correct. Please double check.

3.     The authors did not provide the details for the calculation of adjusted OR.